# SEMI-PARAMETRIC PROMPT-GENERATION FOR MODEL EDITING

## ABSTRACT

Large Language models are used in various downstream tasks with great success. However, changing specific knowledge or beliefs of a model (a.k.a. model editing) efficiently to revise inaccurate predictions while not affecting all other cases is still challenging. Most previous methods compute gradients to change the model. These strategies generally work, paying the cost of high computing and memory complexity. The semi-parametric strategy has recently shown its effectiveness in alleviating the complexity via introducing memory to store the edits of knowledge. However, the memory does not have a proper mechanism to be utilized by a large pre-trained language model, limiting its generalizability to more complicated model editing scenarios. This work proposes a prompt generation mechanism to bridge the gap. Our method encodes the edits as prefix prompts for language models, then has the large pre-trained language model perform inference with the prompts. In other words, the model is edited by prompts without changing model parameters. Our method, SEPROG, significantly outperforms state-of-art methods by up to 20% on entailed edit benchmarks and provides up to 30% better performance over gradient-based methods on non-entailed benchmarks. These advantages are achieved with much less computation and memory consumption, proving prompt generation's great potential in model editing problems.

## 1 INTRODUCTION

Large pre-trained language models (Devlin et al., 2018; Lewis et al., 2019; Radford et al., 2019; Liu et al., 2019) have shown tremendous success on a wide variety of downstream tasks (Brown et al., 2020) such as language generation, fact-checking, summarization, etc. These successes are due to their ability to capture world-scaled knowledge by pre-training on massive corpora (Petroni et al., 2019), as well as their effectiveness in fine-tuning to adapt to arbitrary downstream tasks.

However, modifying the underlying beliefs of large language models with a desired degree of control is still an open problem (Hase et al., 2021). The need to evolve the model's beliefs may range from reflecting simple factual changes about the world (such as changing the capital of a country) to updating entailed relationships between knowledge entities (such as deducing properties of a new species based on its biological taxonomy). The problem setting of *Model Editing* (Mitchell et al., 2021; Sinitsin et al., 2020) formulates the challenge well. Specifically, given a small sample of *edit data* (*e.g.*, description of factual changes), the goal is to make the model provide updated predictions for inputs that are semantically related to the edit data (*i.e.*, *in-scope* data), while retaining the same beliefs for inputs outside the scope of edit data (*i.e.*, *out-scope* data).

Previous model editing strategies learn a gradient-based optimizer (Mitchell et al., 2021) or a model that can quickly adopt the edits via gradient descents (Sinitsin et al., 2020; Hase et al., 2021). These methods achieve significant success with few edits, but their accuracy falls quickly with a larger amount of edits. The interference between edits may be a cause, but controlling the beliefs through the space of model parameters introduces unmanageable complexity. One obvious side-effect is scalability. Mitchell et al. (2021) has shown it is non-trivial to generate the gradients by neural networks or impractical to compute the gradient of gradients (*i.e.*, hypergradients) for learning the fast adapting models beyond billions of parameters.

A recent work, SERAC (Mitchell et al., 2022), tackles model editing using a semi-parametric approach with an explicit memory to store the edit data. SERAC first classifies whether an input is in-scope or

out-scope. If a case is out-scope (*i.e.*, unrelated to the edit data), SERAC uses the base model for predictions. In the in-scope case, SERAC uses an additional model (named counterfactual model in the original paper) for predictions. The counterfactual model extracts the most similar edit in the memory, then uses the edit together with the input for generating predictions. SERAC updates only its memory when there are new edits; therefore, it avoids maneuvering the model parameter space. This strategy is, therefore, scalable and agnostic to the base model. However, the two branches in SERAC make predictions independently, isolating the counterfactual model from leveraging the knowledge of the base model, which was trained with massive corpora. In other words, the design has no knowledge sharing between the branches, making an entailed prediction less likely for the edit-related (in-scope) inputs. This limitation is convoluted with the challenge of training a generalizable classifier to distinguish in-/out-scope, causing the strategy to stumble on harder cases (examined in later sections).

This work proposes SEPROG (Semi-parametric Prompt-Generation) to retain the good parts of the above strategies while minimizing the limitations: (1) scalable base model size; (2) scalable edit dataset size; (3) good generalizability to hard edits (4) low cost to train and inference. Inspired by the non-parametric neural models (Graves et al., 2014; Garnelo et al., 2018a), which stores training data (or edits) rather than model parameters, our semi-parametric approach stores latent representations of the edit data as well as leverages the information from pre-trained base model. Our approach also builds on recent advances in prompt-tuning methods (Li & Liang, 2021; Lester et al., 2021), which convert task descriptions (or the edits) to real-valued prefix for language inputs. These two ideas lead to an end-to-end solution to change the belief of a model with encoder-decoder structures (Lewis et al., 2019; Raffel et al., 2020). Our method generates edit dataset-specific embeddings by leveraging the encoder, then injects the embeddings as the prefix of inputs to the decoder (Figure 1). As a result, SEPROG does not change the base model after deployment (the advantage of SERAC) while still using the base model to generate predictions on both in-scope and out-scope inputs. The inference of in-scope inputs therefore can leverage the world knowledge of the base model to achieve better generalization on hard edits (the advantage of gradient-based methods). Our contributions are summarized as follows:

- We propose a novel model editing method that leverages both prompting and semi-parametric strategies.

- We examined 6 model editing strategies on 3 evaluation aspects: (1) small/large amount of edit data; (2) easy/hard edits; (3) cost of training/inference. This comprehensive comparison shows that the dominant strategy is yet to come, but SEPROG's well-balanced advantages in all 3 aspects make it the best frontier in the current solution space.

## 2 MODEL EDITING PROBLEM

Let $M_{base}$ be a large pre-trained language model trained on a task $\mathcal{T}$ with dataset $\mathcal{D}_\mathcal{T} = \{[x_i, y_i]\}_{i=1}^{|\mathcal{D}_\mathcal{T}|}$. The goal of model editing is to update the beliefs of $M_{base}$ with a set of $K$ *edit descriptors* $Z^e = \{z_i^e\}_{i=1}^K$. In the case of question-answering, the edit descriptor could be a question-answer pair $z_i^e = [x_i^e, y_i^e]$. The edited model will need to provide updated answers for questions that are semantically related to data in $Z^e$ while not changing its predictions for inputs unrelated to $Z^e$.

Formally, we are given a model $M_{base} : \mathbb{R}^n \to \mathbb{R}^m$ trained on dataset $\mathcal{D}_\mathcal{T}$. Let $\mathcal{I}(z^e) = \{[x, y]\}$ be the set of inputs whose predictions are affected by information in edit descriptor $z^e$, i.e., labels $y$ is different for datapoints in $\mathcal{I}(z^e)$ due to updated information from $z^e$. (Note that $z^e \in \mathcal{I}(z^e)$). Let $\mathcal{O}(z^e) \subset \mathcal{D}_\mathcal{T}$ be set of inputs whose predictions are not affected by $z^e$. Then we define $\mathcal{I}(Z^e) = \bigcup_{z^e} \mathcal{I}(z^e)$ as the *in-scope* data and $\mathcal{O}(Z^e) = \bigcup_{z^e} \mathcal{O}(z^e)$ as *out-scope* data.

The goal of model-editing is to derive an edited model $M_{edit}$ from $M_{base}$ such that:

1. $M_{edit}$ outputs the revised predictions for in-scope data ($[x', y'] \sim \mathcal{I}(Z^e)$): $\arg\max_y p(y|x'; M_{edit}) = y'$

2. The changes in the output probability distribution should be small for *out-scope* inputs. In other words, the KL-Divergence between $M_{base}$ and $M_{edit}$'s predictions is small: $\min_{M_{edit}} KL(p(\cdot|x, M_{base})||p(\cdot|x, M_{edit}))$.

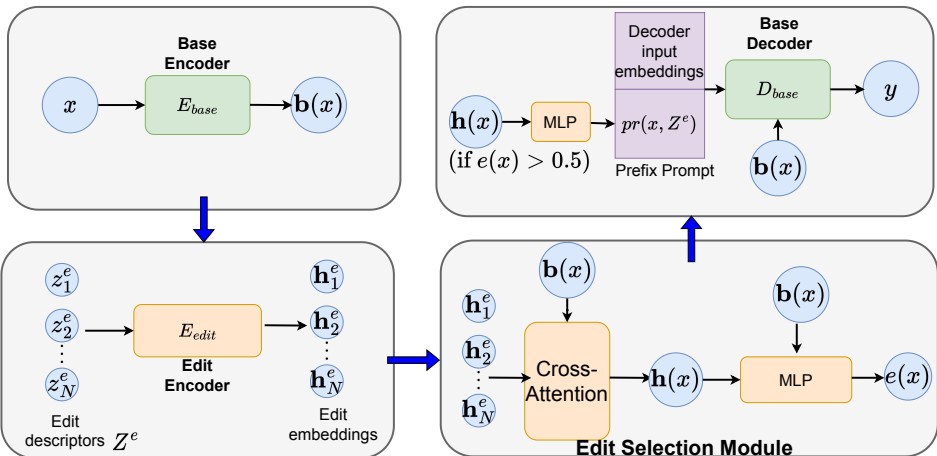

Figure 1: Overview of a version of SEPROG pipeline. The weights of pre-trained base encoder $E_{base}$ and decoder $D_{base}$ are fixed (denoted by Green). SEPROG (denoted by Orange) consists of Edit encoder $E_{edit}$ that encodes edit descriptors, Edit selection module that extracts relevant information related to input $x$ from edit descriptors and classifies $x$ as in-scope or out-scope with probability $e(x)$. If in-scope, SEPROG generates prefix prompts $pr(x, Z^e)$ to concatenate with input to base decoder.

## 3 SEPROG: MODEL EDITING VIA PREFIX PROMPT GENERATION

### 3.1 OVERVIEW

Our approach assumes that the base model $M_{base}$ is an encoder-decoder architecture with encoder $E_{base}$ and decoder $D_{base}$. Many widely used language models fall in this category (Raffel et al., 2020; Lewis et al., 2019; Brown et al., 2020). Unlike most previous works on model-editing that altered the weights of the base model (Mitchell et al., 2021; 2022; Sinitsin et al., 2020), SEPROG trains a neural model to learn to generate prompt embeddings $pr(Z^e, x)$ based on the edit descriptors $Z^e$ and input $x$. The generated prompt $pr(Z^e, x)$ provides information on the parts of the edit descriptors relevant to input so that the model can use the edit information to provide updated beliefs. This prompt is prepended to the token embeddings as input for decoder $D_{base}$ to inform the model to provide updated predictions.

### 3.2 MODEL ARCHITECTURE

SEPROG contains 3 modules to generate the prompt: 1) Edit encoder 2) Edit selection module and 3) Prompt output decoder.

**Edit Encoder** Edit encoder $E_{edit}$ converts each of the edit descriptors $z_i^e$ to a fixed-sized embeddings $\mathbf{h}_i^e$:

$$\mathbf{h}_i^e = E_{edit}(z_i^e) \tag{1}$$

Any encoder style architecture like BERT (Devlin et al., 2018) or encoders of a seq2seq architecture can be used based on the task and the base model.

**Edit selection module** Given the set of embeddings of edit descriptors $H^e = \{\mathbf{h}_i^e\}_{i=1}^N$ and encoding of input $\mathbf{x}$ as $\mathbf{b}(\mathbf{x}) = E_{base}(\mathbf{x})$, edit module first selects the most relevant edit descriptor via cross-attention:

$$\mathbf{h}(\mathbf{x}) = CrossAttention(\mathbf{W}_{att}\mathbf{b}(\mathbf{x}), H^e) \tag{2}$$

where $\mathbf{W}_{att}$ a learnable weight matrix that performs a linear transform on $\mathbf{b}(x)$.

Since the input $x$ maybe either in-scope or out-scope, we also introduce an edit classifier module that predicts if $x$ is in-scope using information from cross-attention layer:

$$e(x) = \sigma\left(NN_e(\mathbf{h}(x) \oplus \mathbf{b}(x))\right) \tag{3}$$

where $NN_e$ is a simple feed-forward neural network with last layer returning a scalar, $\oplus$ is concatenation operator and $\sigma$ is the sigmoid function. Therefore, $e(x) \in [0, 1]$ is the probability of $x$ belonging to in-scope set $\mathcal{I}(Z^e)$.

**Prompt Output Decoder**   Given the relevancy information $\mathbf{h}(x)$ of edit descriptors $Z^e$ to input $x$ as well as likelihood $e(x)$ that $x$ is in-scope, we now leverage this information to generate prompts $pr(x, Z^e) \in \mathbb{R}^{p \times d}$ where $p$ is the number of prompt embeddings and $d$ is the input token embedding dimension of base decoder $D_{base}$.

First note that we only produce the prompts if $e(x) > 0.5$, i.e., the model classifies $x$ as in-scope. We use a feed-forward neural network $NN_{prompt}$ that produces a $p \times d$ dimensional embedding:

$$pr(x, Z^e) = NN_{prompt}(\mathbf{h}(x)). \tag{4}$$

Since the prompt is fed into the unchanged base decoder $D_{base}$, the decoder leverages information from the prompt as well as its prior knowledge from pre-training to effectively provide updated output predictions.

### 3.3   TRAINING THE MODEL

The training of SEPROG learns only the 3 modules that output $h(x)$, $e(x)$, and $pr(x, Z^e)$ with frozen base model ($E_{base}$ and $D_{base}$). The 3 modules are optimized with two losses: scope classification loss and prompt guidance loss.

**Scope classification loss**   The weights of edit encoder $E_{edit}$ and the components of edit selection module ($\mathbf{W}_{att}, CrossAttention$ and $NN_e$) are optimized with this loss to predict the in-scope likelihood $e(x)$:

$$\mathcal{L}_{sc} = -\mathbb{E}_{Z^e} \left[ \mathbb{E}_{[x, \cdot] \in \mathcal{I}(Z^e)} \log e(x) + \mathbb{E}_{[x, \cdot] \in \mathcal{O}(Z^e)} (1 - \log e(x)) \right]. \tag{5}$$

**Prompt Guidance Loss**   All the components of SEPROG used to generate in-scope prompts will be jointly optimized with $\mathcal{L}_{pg}$ to change the output from $D_{base}$:

$$\mathcal{L}_{pg} = -\mathbb{E}_{Z^e} \left[ \mathbb{E}_{[x, y] \in \mathcal{I}(Z^e)} \log p_{D_{base}}(y | pr(x, Z^e), E_{base}(x)) \right]. \tag{6}$$

For each in-scope input $x$, we obtain base encoder output $E_{base}(x)$ and generate prompt $pr(x, Z^e)$ from SEPROG which is fed as prefix to $D_{base}$. The output predictions of $D_{base}$ are optimized towards updated ground-truth $y$ by tuning the parameters of SEPROG. The total training loss is a linear combination of both losses with hyperparameter $\lambda$:

$$\mathcal{L} = \lambda \mathcal{L}_{sc} + (1 - \lambda) \mathcal{L}_{pg} \tag{7}$$

## 4   RELATED WORK

**Model Editing**   Editing model beliefs based on a small set of out-of-distribution samples has been an active area of research with recent advances in the effectiveness of using large pre-trained models on a wide range of NLG and NLU tasks. Most straightforward approaches involve fine-tuning a full set or partial subset of model weights (Zhu et al., 2020). However, these approaches overfit to edit datasets while degrading performance on out-scope data. Sinitsin et al. (2020) propose a gradient-based meta-learning approach based on MAML (Finn et al., 2017) of learning to learn to update model weights for new edit dataset during test-time while retaining performance on out-scope datasets. This method, however, is prohibitively expensive for large models with $\geq 10^7$ parameters. Another alternative line of work (Mitchell et al., 2021; De Cao et al., 2021; Hase et al., 2021) for learning to update model weights instead learn independent smaller neural networks that input gradients of model weights on edit task and provide refined gradients. Since these methods do not retrain model weights, they are relatively scalable but still computationally expensive to train. Moreover, they do not adapt to edit datasets for large sizes. SERAC (Mitchell et al., 2022) propose an alternative gradient-free, model-agnostic approach for editing where they classify each input as related to the edit dataset and allocate out-scope inputs to underlying base models while using a separate counterfactual model for dealing with in-scope inputs. While this method effectively deals with benchmarks that have simple factual edits in edit datasets and similarity between edit dataset and input is easy to detect they fail on more complex scenarios involving complex relations between input and edit such as entailment or leveraging nuanced world knowledge from the base model.

**Neural non-parametric Models** In the general machine learning setup, most models are parametric, i.e., we learn the optimal set of parameters $\theta$ for function $f_\theta$ that learns a mapping $\mathcal{X} \rightarrow \mathcal{Y}$ that fits the data $\mathcal{D} = \{\mathbf{x_i}, \mathbf{y_i}\}_i$. Non-parametric models, in contrast, use the training data to directly derive a functional: $\mathbf{y} = f_\theta(\mathbf{x}_i; \mathcal{D})$. Memory or retrieval based models (Graves et al., 2014; Rastogi et al., 2022; Yogatama et al., 2021) fall into this category.

Neural Process based models (Garnelo et al., 2018a;b) leverage this explicit dependency of the model's functional with the dataset towards multi-task learning and zero-shot meta-learning problems. At a high level, neural process models learn a latent embedding for each task based on the training dataset of the task. This task embedding is in turn used by decoder to provide predictions for test inputs related to the given task. Since these methods explicitly model dependency of training data in the predictive process they are used in various applications where modeling these dependencies improves inductive bias of such models (Rastogi et al., 2022) or provide robust and interpretable predictions (Kamarthi et al., 2021).

**Prompt based fine-tuning** Using human-generated prompts as prefixes to inputs has shown to be an effective method of zero-shot tuning of large-scale models to specific downstream tasks (Brown et al., 2020) without requiring any update to model parameters. However, these approaches are limited by the length of input prompts and the requirement of expert human prompting, which may be sub-optimal for some tasks.

Li & Liang (2021) propose to learn optimal hidden activation prefixes for transformer layers of language models that are fine-tuned for the given downstream task (without changing model weights). They show surprisingly similar performance to explicit fine-tuning while optimizing a very small number of parameters of prefix activations. Lester et al. (2021) instead proposed to optimize only prefix word embeddings for each task showing similar downstream performance with an even lesser number of parameters to optimize. Liu et al. (2021) further generalizes this approach beyond prefixes to have prompts between inputs and output. While prompt tuning methods mostly focus on training prompt embeddings for specific tasks, we study prompt generation to produce prompts specific to the input given the edit descriptors to inform the base model of updated belief from the batch of edit descriptors that are relevant to input.

## 5 EXPERIMENT SETUP

### 5.1 DATASETS

Table 1: Examples from Copy datasets (zsRE, FEVER, Wikipedia) and Entailed Datasets (LeapOfThough, Wikidata5m).

| Dataset | Data Type | Input | Label |
|---|---|---|---|
| zsRE | Edit Descriptor | Who is the Sun Public License named after? | Sun Micro Devices |
| | In-Scope | The Sun Public License has been named for whom? | Sun Micro Devices |
| | Out-Scope | What continent is Mount Whillans found on? | Antartica |
| FEVER | Edit Descriptor | In 1985, Cyndi Lauper won the best New Artist Award at the 27th Grammy Awards. | False |
| | In-Scope | At the 27th Grammy Awards in 1985 Cyndi Lauper won the Best New Artist Award. | False |
| | Out-Scope | Tetris has sold millions of copies. | False |
| Wikipedia Text Generation | Edit Descriptor (Same as in-scope) | Du Fu's mother died shortly after he was born, and he was partially raised by his aunt.... | - |
| | Out-Scope | While scientific experiments performed by Clementine and Lunar Prospector could indicate the presence of water... | - |
| LeapOfThought | Edit Descriptor | A viper is a vertebrate. | True |
| | Simple In-Scope | Viper is an example of a vertebrate. | True |
| | Entailed In-Scope | A viper has a brain. | True |
| | Out-Scope | A Goldfish has a fin. | True |
| Wikidata5m | Edit Descriptor | Mary Good has relation 'award received' to | Garvan-Olin Medal |
| | Simple In-Scope | Mary Good has relation 'winner of' to | Garvan-Olin Medal |
| | Hard Out-Scope | Mary Good has relation 'educated at' to | U Arkansas |

We experiment with two types of datasets: **Copy-edit datasets** (easy edits) and **Entail-edit datasets** (hard edits), each of which are shown to be effective with memory-based (Mitchell et al., 2022) and gradient-based methods (Mitchell et al., 2021; Sinitsin et al., 2020), respectively.

**Copy-edit datasets** These datasets are designed such that the inputs of in-scope datapoint $[x', y']$ have very similar input $x'$ (rephrases) to the input $x^e$ of edit descriptors $[x^e, z^e]$ and the output ground truth of these in-scope examples are identical to the output of edit descriptor ($y' = y^e$). In other words, Copy-edit datasets require $M_{edit}$ to simply map the in-scope input to the relevant edit descriptor and copy the label from the edit-descriptor. We include the following 3 copy-edit datasets:

1. *zsRE Question Answering*: It contains 151631 questions based on factual knowledge from Wikipedia (Levy et al., 2017). We use the train/validation/test split similar to De Cao et al. (2021). The in-scope inputs are rephrased questions with the same answer.

2. *FEVER Fact-checking* (Thorne et al., 2018): It contains 115409 factual claims with True/False labels. In-scope data include rephrases of updated fact $x^e$ in the edit-descriptor.

3. *Wikipedia Text Generation*: For each edit descriptor input $x^e$ taken from WikiText-103, an alternate 10-token completion generated from a pre-trained distilGPT-2 model is chosen as output. There are no alternate in-scope inputs; therefore, this tests whether the model can replicate $y^e$.

**Entail-edit datasets** These datasets were chosen such that in-scope examples have complex entailment relationships with edit descriptors and require retaining and leveraging additional knowledge from beliefs of the base model. These benchmarks may include hard out-scope data related to edit descriptors but with a different label. We evaluate on following 2 datasets used in Hase et al. (2021):

1. *LeapOfThought (LoT):* This dataset introduced by Talmor et al. (2020) consists of claims labeled as true or false based on supporting facts.

2. *Wikidata5m:* This benchmark uses a relational database (Wang et al., 2021) similar to Hase et al. (2021) to build this dataset. The database contains triplets $(o_1, r, o_2)$ where $o_1$ and $o_2$ are objects with relation $r$. In general cases, a triplet $(o_1, r, o_2)$ is not related to triplet $(o_1, r', o_3)$; therefore, having $(o_1, r, o_2)$ as an edit should not affect the prediction of $(o_1, r', o_3)$, unless $r'$ is a paraphrase of $r$. This property allows the benchmark to test $M_{edit}$'s ability in discriminating the hard out-scope inputs (with unrelated $r'$) from in-scope inputs (with paraphrased $r$).

We provide examples from each dataset in Table 1.

## 5.2 BASELINE METHODS

We includes four gradient-based (1 to 4) and one memory-based methods. Each of them may target different aspects of model editing problem and have specific strength. We compare all of them under a coherent experimental setting. These methods are:

1. FT: It directly fine-tunes the model with edit descriptor using the total loss $\mathcal{L}$ (Equation 7).

2. ENN (Sinitsin et al., 2020): Editable Neural Networks (ENN) uses a bi-level optimization meta-learning similar to MAML (Finn et al., 2017) to make the base model adaptable to simple fine-tuning with edit descriptors.

3. SLAG (Hase et al., 2021): It extends the learned optimizer method from De Cao et al. (2021) for a larger amount of edits and entailed edits.

4. MEND (Mitchell et al., 2021): It is a state-of-art method that learns a special neural networks to generate the gradient update for specific layers of the base model during fine-tuning.

5. SERAC (Mitchell et al., 2022): The state-of-art memory-based method performs particularly effectively on copy-edit datasets.

## 5.3 BACKBONE MODEL AND TASK-SPECIFIC ARCHITECTURE

For base models, similar to Mitchell et al. (2022; 2021), we use pre-trained T5 model (Raffel et al., 2020) (`google/t5-large-ssm-nq`) for ZSRE, BERT (Devlin et al., 2018) (`bert-base-uncased`) for FEVER and DistilGPT2 (Sanh et al., 2019) for Wikipedia generation. Similar to Hase et al. (2021) we use BART (Lewis et al., 2019) (`facebook/bart-base`)

Table 2: Average Edit Success (ES↑) and Drawdown (DD↓) of SEPROG and baselines. Each value is the averaged results from edit batch size $\{1, 4, 10, 20, 64, 128\}$. The Avg. column, which is the average of all 5 datasets.

| | | | Copy-edit dataset | | | | | | Entail-edit dataset | | | |
| | Avg. | | ZSRE | | FEVER | | Wikipedia | | LoT | | Wikidata5m | |
| Method | ES | DD | ES | DD | ES | DD | ES | DD | ES | DD | ES | DD |
|---|---|---|---|---|---|---|---|---|---|---|---|---|
| FT | 0.58 | 0.38 | 0.65 | 0.47 | 0.82 | 0.15 | 0.46 | 0.36 | 0.49 | 0.45 | 0.50 | 0.47 |
| ENN | 0.67 | 0.27 | 0.72 | 0.31 | 0.86 | 0.13 | 0.56 | 0.25 | 0.63 | 0.37 | 0.59 | 0.27 |
| SLAG | 0.69 | 0.33 | 0.77 | 0.30 | 0.87 | 0.17 | 0.72 | 0.27 | 0.49 | 0.43 | 0.61 | 0.46 |
| MEND | 0.73 | 0.24 | 0.68 | 0.18 | 0.89 | 0.07 | 0.71 | 0.26 | 0.67 | 0.33 | **0.68** | 0.34 |
| SERAC | 0.76 | **0.15** | **0.95** | **0.07** | **0.96** | **0.05** | **0.97** | **0.18** | 0.43 | **0.22** | 0.47 | 0.24 |
| SEPROG | **0.78** | 0.18 | 0.78 | 0.18 | 0.91 | 0.06 | 0.83 | 0.23 | **0.72** | 0.26 | 0.67 | **0.18** |

for LeapOfThought and Wikidata5m. We edit only the last two layers of the backbone models' encoders and decoders for all gradient-based methods. For both ZSRE and Wikipedia generation, we use `t5-small` for the edit encoder due to its small computational complexity and memory consumption. We use `bert-base-uncased` for FEVER and `facebook/bart-base` encoder for LeapOfThought and Wikidata5m.

## 5.4 EVALUATION METRICS

Following prior works (Sinitsin et al., 2020; De Cao et al., 2021; Hase et al., 2021; Mitchell et al., 2021; 2022), we use Edit Success (ES) and Drawdown (DD) as evaluation metrics.

**Edit Success (ES)** measures the ability of the model to change its predictions on in-scope inputs based on edit descriptors. For a model $M_{edit}$ with edit descriptors $Z^e$, the edit success is defined as the average times the prediction of the model is an exact match to the ground truth of in-scope input:

$$ES_{M_{edit}}(Z^e) = \mathbb{E}_{[x,y] \in \mathcal{I}(Z^e)} \mathbf{1}\{M_{edit}(x) = y\} \tag{8}$$

**Drawdown(DD)** measures model's ability to retain similar predictions for out-scope inputs. DD is defined as:

$$DD_{M_{edit}}(Z^e) = \mathbb{E}_{[x,y] \in \mathcal{O}(Z^e)} \mathbf{1}\{M_{edit}(x) \neq y\} \tag{9}$$

## 6 RESULTS

## 6.1 MODEL EDITING PERFORMANCE

To analyze the performance of SEPROG and baselines on edit batches of varying sizes, we evaluate for batch sizes in $\{1, 4, 10, 20, 64, 128\}$. The average Edit Success (ES) and Drawdown (DD) are reported in Table 2. On copy-edit datasets, SERAC significantly outperforms other models in both ES and DD due to the easier task of classifying in-scope and out-scope data and copying labels of relevant edit descriptors for in-scope inputs. In the case of entail-edit datasets, on average (over all batch sizes) SEPROG outperforms baselines by 1-7% in ES and has comparable drawdown to best performing models. We will now look at the performance of models across varying edit batch sizes as shown in Figure 2.

**Performance on copy-edit datasets across batch size** SERAC consistently outperforms all other models across varying batch sizes due reasons discussed above. However, we observe that SEPROG consistently outperforms all other gradient-based baselines in both ES and DD for larger edit batch sizes (64 and 128) with 7-30% better ES and 14-20% DD. This is due to our approach of generating prompts based on the automatic selection of relevant edit descriptors and providing these prompts as prefixes to the decoder without changing the weights of the base model.

**Performance on entail-edit dataset across batch sizes** Unlike the previous case, SERAC's performance is poor in entail-edit datasets that require knowledge from the base model to deal with entailed predictions for in-scope inputs. For larger batches (64 and 128), SEPROG outperforms all

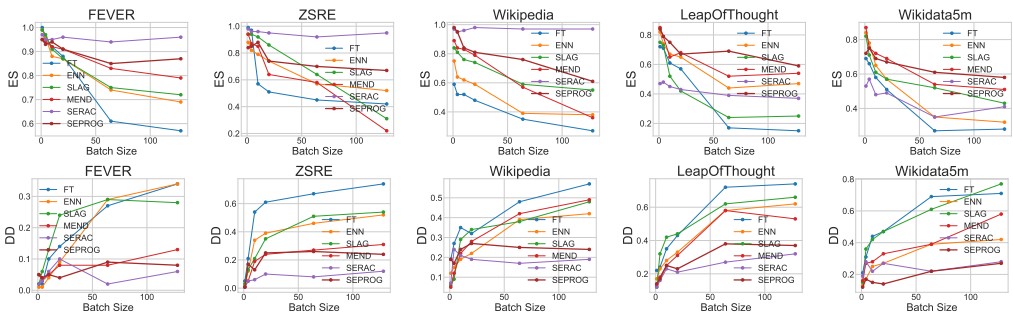

Figure 2: *ES and DD of* SEPROG *and baselines across varying edit batch sizes.*

baselines in ES with 13-20% better scores. Meanwhile, SEPROG's DD performance is comparable to SERAC, both of which outperform gradient-based baselines. Therefore, due to the combination of separate edit descriptor retrieval mechanism as well as leveraging information from base model, SEPROG overcomes shortcomings of both memory-based and gradient-based baselines to provide state-of-art performance on entail-edit datasets for large edit batches.

## 6.2 EFFICIENCY OF SEPROG

A major issue with the deployment of gradient-based methods is their large compute and memory requirements, making it prohibitively expensive for editing large language models. For example, ENN needs to compute hypergradients on model weights during the training of the base model. While methods like MEND and SLAG use a separate set of neural networks to modify gradient updates of the base model, computing the hypergradients to train these neural networks is still expensive and consumes much memory.

Memory-based methods, like SERAC and our SEPROG, act as wrappers around the base model without updating the base model's weights. Moreover, while SERAC trains a separate counterfactual model to deal with in-scope inputs, SEPROG only learns to generate prefix prompts and does not need to train another language model. However, SEPROG needs to calculate gradients of the decoder model to derive the intermediate gradients $\nabla_{pr(x,Z^e)}\mathcal{L}$ w.r.t the input prompt.

Table 3: *Average Wall time and peak GPU memory use during training and inference loop for a single batch of size 64.*

| | Training | | Inference | |
|---|---|---|---|---|
| Models | Time (ms.) | Memory (GB) | Time (ms.) | Memory |
| FT | NA | NA | 1262 | 8.9 |
| ENN | 8948 | 22 | 653 | 4.8 |
| SLAG | 2157 | 9.2 | 629 | 5.2 |
| MEND | 1755 | 9.4 | 517 | 5.5 |
| SERAC | 235 | 3.6 | 57 | 3.4 |
| SEPROG | 127 | 4.6 | 84 | 2.9 |

To measure the compute and memory requirement during editing of a large language model, we measure the time it takes and the peak GPU memory usage during training and inference for a single batch of size 64 of ZSRE dataset (with T5 base model) on an NVIDIA Tesla V100 GPU with 32 GB VRAM in Table 3 (Results for additional batch sizes are in Appendix Section C). During training, we observe that the time and memory requirements of SEPROG are around 3 times and 2 times lesser than the most efficient gradient-based baselines, respectively. SERAC is observed to be 1.8 and 1.2 times more efficient than SEPROG for compute time and memory usage respectively. However, during inference, SEPROG uses the least memory since it does not need to calculate gradients like gradient-based methods or use two separate language models for in-scope and out-scope inputs like SERAC.

## 6.3 ABLATION STUDIES

**Ablation variants of SEPROG**    We study the effectiveness of prompt-generating without fine-tuning model weights as well as the efficacy of edit classifier modules using the following variants of SEPROG:

- SP-FINETUNE: We use generated prefixes from SEPROG and fine-tune the base model.

- SP-NOCLASSIFIER: We remove the edit classifier (Equation 3) and do not use Scope Classification Loss.

- SP-NOSELECT: We have scope classification during training but remove it in inference. In other words, the generated prefix will always inject into the decoder, no matter whether the input is in-scope or out-scope.

Prefix-tuning has shown to be an efficient method of tuning a large language model Li & Liang (2021). Moreover, Li & Liang (2021) and Lester et al. (2021) showed that they are more generalizable to domain transfers and don't overfit to training data. Therefore, SP-FINETUNE is designed to test the hypothesis that using prompt generation alone is a more effective method of preventing overfitting and handling large edit batches than updating model weights on new datasets. We also test the efficacy of having a separate classifier module to avoid interference between in-scope and out-scope inputs during training and inference through SP-NOCLASSIFIER and SP-NOSELECT, respectively.

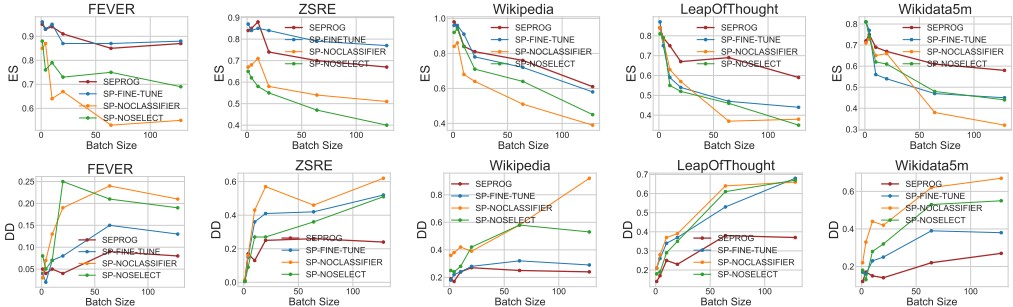

Figure 3: *ES and DD of* SEPROG *and ablation variants across varying edit batch sizes.*

**Edit performance of the ablation variants**    We compare ES and DD for all edit batch sizes in Figure 3. On average, we observe that SEPROG outperforms the best performing variant in both ES and DD by 5% and 31%, respectively. For larger batch sizes of 64 and 128, we observe 8% better ES and 24% better DD on average. SP-NOCLASSIFIER is the worst performing model in terms of DD, followed by SP-NOSELECT, showing the efficacy of the edit classifier module both for tuning the modules of SEPROG via additional signal for learning supervision as well as for selecting when to append prefix prompts.

## 7 CONCLUSION

We proposed SEPROG, a prefix generation model for effective model editing over large edit batch sizes. SEPROG overcomes the scalability issues of gradient-based methods and inflexibility of memory-based approaches by directly leveraging the base model for adapting to edit dataset by providing appropriate prefix prompts based on the relevance of input to edit dataset. We observed up to 30% improvement in edit performance over gradient-based models on copy-edit datasets and up to 20% better scores than all state-of-art models for harder entail-edit datasets with large edit batches. Due to our prefix generation model, learning over a much smaller number of parameters SEPROG is also 2-3 times more efficient than gradient-based approaches.

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

# Appendix for Semi-parametric Prompt-Generation for Model Editing

## A  DATASET AND BASELINE CODE

The copy-edit datasets were sourced from Mitchell et al. (2021) [1]. LeapOfThought dataset was constructed based on code from Hase et al. (2021)[2]. We also use the same train/validation/test splits as in the official data sources.

We use the code for all gradient-based methods except SLAG from the official code of Mitchell et al. (2021)[1]. We used official implementation of SLAG from the authors Hase et al. (2021)[2]. Since the official implementation for SERAC is not publicly available at the time of writing, we implemented it based on details from Mitchell et al. (2022) and observed the results to be similar to that reported in the paper.

## B  HYPERPARAMETERS

**Architecture Details**   We appended a single feed-forward layer of 768 units to all edit encoders such that $\mathbf{h}_i^e \in \mathbb{R}^{768}$. The learnable weight $\mathbf{W}_{att}$ of $CrossAttention$ has dimensions $768 \times d$ where $d$ is the output dimensions of $\mathbf{b}(x)$ from base encoder $E_{base}$. $NN_e$ is a two-layer feed-forward with all hidden and output layers having 768 units. While larger prefix-length $p$ generally provides better performance (Li & Liang, 2021), we found the marginal performance gains to be minimal for $p$ greater than 32 in all benchmarks.

**Training specifics**   In all benchmarks, we used an Adam optimizer (Kingma & Ba, 2014) with a linear warm-up in learning rate over the first 200 epochs and gradient clipping threshold set at 5.0. We use early stopping with the patience of 200 epochs over total loss on the validation data for copy-edit datasets and 1000 epochs for entail-edit datasets. For each benchmark, we mostly tuned over learning rate and $\lambda$ and found that the learning rate between 1e-5 and 1e-4 and $\lambda \in [0.1, 0.5]$ provided good performance. During training, we sample the same number of in-scope and out-scope samples per batch and we use only in-scope examples to derive prompt guidance loss whereas we use both for scope classification loss.

## C  TRAINING AND INFERENCE TIMES FOR VARYING BATCH SIZES

Table 4: *Average Wall time and peak GPU memory use during training and inference loop for a single batch of size 32.*

| Models | Training | | Inference | |
|---|---|---|---|---|
| | Time (ms.) | Memory (GB) | Time (ms.) | Memory |
| FT | NA | NA | 922 | 6.8 |
| ENN | 7229.5 | 18.3 | 419 | 4.1 |
| SLAG | 5883 | 6.95 | 495 | 4.2 |
| MEND | 3108 | 7.65 | 351 | 5.1 |
| SERAC | 194 | 3.3 | 48 | 3.3 |
| SePropTe | 109 | 3.8 | 69 | 2.8 |

---

[1] https://github.com/eric-mitchell/mend
[2] https://github.com/peterbhase/SLAG-Belief-Updating

Table 5: *Average Wall time and peak GPU memory use during training and inference loop for a single batch of size 64.*

| Models | Training | | Inference | |
|--------|----------|---|-----------|---|
| | Time (ms.) | Memory (GB) | Time (ms.) | Memory |
| FT | NA | NA | 1262 | 8.9 |
| ENN | 8948 | 22 | 653 | 4.8 |
| SLAG | 2157 | 9.2 | 629 | 5.2 |
| MEND | 1755 | 9.4 | 517 | 5.5 |
| SERAC | 235 | 3.6 | 57 | 3.4 |
| SEPROG | 127 | 4.6 | 84 | 2.9 |

Table 6: *Average Wall time and peak GPU memory use during training and inference loop for a single batch of size 128.*

| Models | Training | | Inference | |
|--------|----------|---|-----------|---|
| | Time (ms.) | Memory (GB) | Time (ms.) | Memory |
| FT | NA | NA | 1955 | 14.6 |
| ENN | 12385 | 29.4 | 1056 | 6.9 |
| SLAG | 7391 | 13.7 | 897 | 7.2 |
| MEND | 5482 | 12.9 | 840 | 6.4 |
| SERAC | 317 | 4.2 | 69 | 3.7 |
| SePropTe | 169 | 6.5 | 103 | 3.1 |

