# OpenReview forum: "Semi-parametric Prompt-Generation for Model Editing"
_ICLR.cc/2023/Conference — Submitted to ICLR 2023_

### Official Review · Reviewer_k5Q7 · 2022-10-21

**Confidence:** 3
**Correctness:** 3
**Technical Novelty And Significance:** 3
**Empirical Novelty And Significance:** 2
**Recommendation:** 5

**Clarity, Quality, Novelty And Reproducibility:**

- The paper could benefit a lot from more editing. Details are missing or confusing to readers (see weaknesses section).
- The idea is novel and easy to follow, it should not be hard to reproduce the results from my experience.

**Strength And Weaknesses:**

### Strength
- The approach is simple and easy to follow. The method visualization is clear.
- I like that the authors did a thorough analysis to understand the training and inference efficiency of the baselines and the proposed approach. This is extremely useful for understanding the practicability of the approaches.

### Weaknesses
- The writing is not very clear, and I have a bunch of questions throughout reading the paper.
    - The concept of entailed predictions is explained in the experiment section, but the concept starts to be used in the introduction. It causes a bit of misunderstanding.
    - The authors should provide better justifications for the selection of the base models. Is it because different tasks require different base encoders? Or simply for a fair comparison to previous works? And why can we not simply use one structure for the framework? The necessity of using different kinds of encoders for different tasks hurts the generalization ability of the approach.
    - Why is performance so sensitive to batch size? Does one batch always contain examples of one edit description or could be any in-scope and out-of-scope examples? Why are the proposed approach and baselines behave differently for Copy-edit datasets and Entail-edit datasets in terms of a change of batch size?
    - Table 2 caption is confusing. Is the average performance averaged over both datasets and batch size, or averaged over datasets on one batch size?
- The results in Table 2 and Table 3 show that the approach is comparable to SERAC in terms of performance and training/inference cost but not significantly better.


**Summary Of The Paper:**

The paper aims to solve the model editing problem via training prefix prompts and using them together with a frozen language model for inference. The proposed approach, SEPROG, is less computationally heady and requires less memory than gradient-based approaches. SEPROG outperforms state-of-the-art methods by 20% on entailed edit benchmarks and provides up to 30% better performance over gradient-based methods on non-entailed benchmarks.

**Summary Of The Review:**

The proposed idea to address the model editing problem is easy and simple to follow. The experimental results are not compelling enough compared to existing approaches.

---

> ### Author Response · Authors · 2022-11-16
> **Reply to Reviewer k5Q7 (Part 1/2)**
>
> We thank the reviewer for their valuable comments. We appreciate the reviewer for recognizing the clarity of presenting our methodology and our detailed evaluation.
>
> We will address the comments and questions of the reviewer.
>
> **The concept of entailed predictions is explained in the experiment section, but the concept starts to be used in the introduction. It causes a bit of misunderstanding.**
>
> Thank you for the feedback. We have mentioned updating entailment relationships of entities as an application of model editing and noted that such complex entailments are harder to handle for previous memory-based methods like SERAC.
>
>
> **The authors should provide better justifications for the selection of the base models. Is it because different tasks require different base encoders? Or simply for a fair comparison to previous works? And why can we not simply use one structure for the framework? The necessity of using different kinds of encoders for different tasks hurts the generalization ability of the approach.**
>
> We use the base models as used in previous works for a fair comparison. Our method makes no assumption about the underlying model structure. The same encoder can be used in different tasks.
>
>
> **Why is performance so sensitive to batch size? Does one batch always contain examples of one edit description or could be any in-scope and out-of-scope examples? Why are the proposed approach and baselines behave differently for Copy-edit datasets and Entail-edit datasets in terms of a change of batch size?**
>
> The edits in a batch may interfere with each other; thus, the batch size is the primary factor to affect the results of all methods. A better method should show a stronger robustness against this factor. Entail-edit is harder than Copy-edit; therefore, all methods show a faster performance drop with a larger edit batch size.
>
> For details, edit batch size refers to the number of edit descriptors. The performance of gradient-based approaches is very sensitive to the batch size since these methods need to modify the gradients (and as a result the weights) of the given base model to encompass the modified beliefs from all the edit descriptors.
>
> For each edit descriptor sampled in a batch, we sample an input sequence that could be in-scope or out-scope (chosen randomly) to train the full model.
>
> As explained in Section 5.1, copy-edit datasets have in-scope examples very similar to edit descriptors which makes scope classification a simple task. Moreover, the ground-truth output of the in-scope data points is contained in the edit descriptors. Therefore, memory-based models like SERAC can efficiently handle large batches of copy-edit datasets and easily learn to copy the relevant output from the edit descriptors.
>
> In the case of entail-edit datasets, scope classification is non-trivial and prediction of ground truth requires background knowledge from the pre-trained base model. Hence SERAC's performance, specifically Edit Success (ES) is much poorer compared to all the gradient-based baselines which leverage the background knowledge from the base models.
>
> To summarize, gradient-based methods can not provide consistent performance over larger edit batch sizes since they directly deal with modifying the base model's parameters. Semi-parametric memory-based models like SERAC, while providing consistent performance over varying batch sizes, do not perform well on entail-edit data since they do not leverage background knowledge of the base models to deal with non-trivial scope classification and prediction task of entail-edit datasets that require this background knowledge.
>
>
> **Table 2 caption is confusing. Is the average performance averaged over both datasets and batch size, or averaged over datasets on one batch size?**
>
> Table 2 reports the average performance over all batch sizes in $\{1,4,10,20,64,128\}$. We report this average performance for each of the datasets as well as the average over all data sets in the *Avg.* column.

---

> > ### Author Response · Authors · 2022-11-16
> > **Reply to Reviewer k5Q7 (Part 2/2)**
> >
> > **The results in Table 2 and Table 3 show that the approach is comparable to SERAC in terms of performance and training/inference cost but not significantly better.**
> >
> > Table 2 shows previous SOTA heavily biased their accuracy to certain types of editing problems while performing badly with the other type. Our method provides SOTA-comparable results in all cases. We consider this a significant advance for the model editing problem.
> >
> > Specifically, SEPROG performs as well as SERAC on copy-edit datasets (where gradient-based methods suffer at large batch sizes) and performs as well as MEND (gradient-based SOTA method) on entail-edit datasets where SERAC’s Edit Success (ES) is poor. Moreover,  for entail-edit datasets with large edit batch sizes (64 and 128), SEPROG clearly outperforms all the baselines by 13-20%.
> >
> > Table 3 shows that our model's computational efficiency and memory requirement is comparable to SERAC even though we leverage information from the base model to provide superior performance to SERAC on entail-edit datasets.

---

### Official Review · Reviewer_8Z3Z · 2022-10-25

**Confidence:** 3
**Clarity, Quality, Novelty And Reproducibility:** see above
**Correctness:** 2
**Technical Novelty And Significance:** 1
**Empirical Novelty And Significance:** 2
**Recommendation:** 3

**Strength And Weaknesses:**

Strength
- Model editing is an important and high-impact problem, which is the focus of this paper.

- The use of prompted language models as counterfactual models is an interesting research direction.

Weakness


- This is a minor extension or a special case of SERAC and there’s no overwhelming evidence that this is better than the original model. (See table 2 & 3)


- The writing can be improved. The paper assumes the reader has detailed knowledge of SERAC and the entire literature on model editing. In order to understand what this paper is about, I have read the SERAC paper.
In order to reach a broader audience, the authors should motivate the model editing problem in the introduction and then clearly explain SERAC and its limitations in the second part.


Some other specific examples related clarity.

- The sentence“ Good parts of the above strategies while minimizing its limitations” is vague: what  are the good parts, and what are the limitations? How is this paper addressing the limitations?

- How is this a “semi-parametric’ model? This was never defined?

- Figure 2 and 3 are near impossible to read

- Table 3 might be interpreted too generously. I would not say that SEPROG is twice as fast as SERAC looking at it. The table also only reports the results for batch size 64, and not the others.

**Summary Of The Paper:**

This paper provides a method for model editing, how to locally update the output of the model to exhibit desirable properties.
It builds on SERAC, which consists of three parts:  cache edits, an edit scope classifier, and a counterfactual model that overrides the base language model. A scope classifier determines when the output of the base language model will be in scope, and a counterfactual model determines what should be edited.
This paper argues that having separate counterfactual and base models is inefficient, since we want to share knowledge between them.
To address this problem, this paper proposes to use the prompted base model as the “counterfactual model”.
The main claim of the paper is that doing so will achieve better generalization on hard edits.
Unfortunately, the experiments do not support the claim.

**Summary Of The Review:**

This paper presents an extension to SERAC, a model editing algorithm.  Unfortunately, the extension is neither theoretically justified nor empirically significant.

---

> ### Author Response · Authors · 2022-11-16
> **Reply to Reviewer 8Z3Z (Part 1/2)**
>
> We thank the reviewer for their valuable comments. We appreciate that the reviewer has recognized the importance of our research direction. We wish to address the following comments and misconceptions:
>
> **This is a minor extension or a special case of SERAC and there’s no overwhelming evidence that this is better than the original model. (See table 2 & 3)**
>
> Our work is the first to introduce a prompt generation mechanism into the model editing problem. The prompt generation is an uncommon strategy even outside the scope of model editing. Most other prompt-related methods optimize the prompt via training, but our method learns to generate the prompt, thus requiring no training to make an edit take effect after a model is deployed. Therefore, our method is not a minor extension of SERAC.
>
> Regarding the evidence, Table 2 shows previous SOTA heavily biased their accuracy to certain types of editing problems while performing badly with the other type. Our method provides SOTA-comparable results in all cases. We consider this a significant advance for the model editing problem.
>
> Specifically, SEPROG performs as well as SERAC on copy-edit datasets (where gradient-based methods suffer at large batch sizes) and performs as well as MEND (gradient-based SOTA method) on entail-edit datasets where SERAC’s Edit Success (ES) is poor. Moreover,  for entail-edit datasets with large edit batch sizes (64 and 128), SEPROG clearly outperforms all the baselines by 13-20%.
>
> **The writing can be improved. The paper assumes the reader has detailed knowledge of SERAC and the entire literature on model editing.**
>
> We have provided a formal definition of the model editing problem in Section 2. We have also contextualized our work against important previous works in model editing along with recent works on prompt-based models. We do not believe that a detailed understanding of SERAC is required to understand our work. While we borrow ideas from SERAC in using a classification module to get relevant edit descriptors, as stated above our approach has important methodological differences that is not related to SERAC.
>
> **The sentence“ Good parts of the above strategies while minimizing its limitations” is vague: what are the good parts, and what are the limitations? How is this paper addressing the limitations?**
>
> We describe the advantages and limitations of gradient-based methods and semi-parametric methods like SERAC. Briefly, gradient-based methods are computationally expensive and hard to scale for larger models and for larger edit batch sizes. However, they directly modify the weights of the model and can leverage the base model’s world-knowledge to deal with harder edit-input data points such as in the case of entail-edit datasets. SERAC, on the other hand, can easily scale to the base model of large sizes and large edit batches but are not effective with complex edit datasets.
>
> SEPROG combines the best of both approaches: adapting to complex edit datasets and larger batch sizes while minimizing their limitations: efficiency and base-model agnostic inference for edit-related inputs.
>
> We deal with the computational limitations as well as leverage base-model information by formulating the problem as a prompt generation task by learning relevancy and information from edit descriptors. The above description was highlighted in the last paragraph of Section 1.
>
> **How is this a “semi-parametric’ model? This was never defined?**
>
> We call our model semi-parametric since it has similarities to non-parametric memory-based models like [1,2,3] which directly leverage the similarity of input with training data points.
> Our work (and SERAC) instead leverage similarity with given input descriptors but also use background information from pre-trained base-model and edit encoder for prediction. We have clarified this in the introduction of the revised submission.
>
> **Figure 2 and 3 are near impossible to read**
>
> We have increased the font size of the figures in the revised submission to improve readability.
>
> **Table 3 might be interpreted too generously. I would not say that SEPROG is twice as fast as SERAC looking at it. The table also only reports the results for batch size 64, and not the others.**
>
> We claim that during training
> >SERAC is observed to be two times more efficient than SEPROG…
>
> (note that SERAC and SEPROG are interchanged compared to the reviewer’s comment) which is actually too generous in favor of SERAC. We revised it to
> >SERAC is observed to be 1.8 and 1.2 times more efficient than SEPROG for compute time and memory usage respectively.
>
> We have also added a similar analysis as Table 3 for batch sizes 32 and 128 in the Appendix of the revised submission.

---

> > ### Author Response · Authors · 2022-11-16
> > **Reply to Reviewer 8Z3Z (Part 2/2)**
> >
> > ### References
> >
> > [1] Alex Graves, Greg Wayne, and Ivo Danihelka. Neural turing machines. arXiv preprint arXiv:1410.5401, 2014
> >
> > [2] Richa Rastogi, Yuntian Deng, Ian Lee, Mert R Sabuncu, and Volodymyr Kuleshov. Semi-parametric deep neural networks in linear time and memory. arXiv preprint arXiv:2205.11718, 2022
> >
> > [3] Dani Yogatama, Cyprien de Masson d’Autume, and Lingpeng Kong. Adaptive semiparametric language models. Transactions of the Association for Computational Linguistics,2021

---

### Official Review · Reviewer_EG39 · 2022-10-29

**Confidence:** 3
**Correctness:** 3
**Technical Novelty And Significance:** 2
**Empirical Novelty And Significance:** 2
**Recommendation:** 3

**Clarity, Quality, Novelty And Reproducibility:**

Overall, the paper is well-written and easy to follow. The method lacks novelty, as it simply uses neural networks to predict in-scope and out-scope data and generating the prompts, and I feel difficult to draw insights from the results.

**Strength And Weaknesses:**

The proposed method is intuitive and utilizes the power of neural networks to generate good prompts. However, the performance is often not better than SERAC, as shown in Table 2. The success of the end-to-end system depends on the correct prediction from the edit classifier module, as well as the effectiveness of the prompt output decoder, which seems quite challenging to me. I expect that to be effective, the SEPROG modules require additional training every time when we want to edit a new set of facts, which makes it less flexible.

**Summary Of The Paper:**

This paper proposes a semi-parametric approach which trains neural network to recognize in-scope and out-scope data, and generating prompts that enables editing the response of the base model. The result is on par with state-of-the-art in most cases, while the training time and inference memory seem better.

**Summary Of The Review:**

I feel model editing is an interesting problem to study, but the proposed method lack novelty, and the result did not show significant improvements over existing approaches. I look forward to the improvements in the next version.

---

> ### Author Response · Authors · 2022-11-16
> **Reply to Reviewer EG39**
>
> We thank the reviewer for their valuable comments. We wish to address the following comments and misconceptions:
>
> **The success of the end-to-end system depends on the correct prediction from the edit classifier module, as well as the effectiveness of the prompt output decoder, which seems quite challenging to me**
>
> Thank you for acknowledging the challenges in this problem setting. Our method addressed both challenges and achieved SOTA-comparable results with a novel prompt-generation strategy. We believe this is a significant advancement for this direction that is worth sharing with the community.
>
>
> **I expect that to be effective, the SEPROG modules require additional training every time when we want to edit a new set of facts, which makes it less flexible.**
>
> This is not true. Once SEPROG is trained on the base model, during deployment on new edit facts, no training is required. At runtime, we perform a forward pass on SEPROG with a new input sequence and new edit data to generate appropriate prompts to modify the predictions of the base model according to the new edit data.
>
>
> **the proposed method lack novelty**
>
> We respectfully disagree both with regard to novelty in terms of empirical results contribution to model editing problem as well as with regard to modeling approach.
> Our work is the first to introduce a prompt generation mechanism into the model editing problem. The prompt generation is an uncommon strategy even outside the scope of model editing. Most other prompt-related methods optimize the prompt via training, but our method learns to generate the prompt, thus requiring no training to make an edit take effect after a model is deployed. This idea hasn’t been explored by any previous work and is highly novel.
>
> **the result did not show significant improvements over existing approaches.**
>
> Table 2 shows previous SOTA heavily biased their accuracy to certain types of editing problems while performing badly with the other type. Our method provides SOTA-comparable results in all cases. We consider this a significant advance for the model editing problem.
>
> Specifically, SEPROG performs as well as SERAC on copy-edit datasets (where gradient-based methods suffer at large batch sizes) and performs as well as MEND (gradient-based SOTA method) on entail-edit datasets where SERAC’s Edit Success (ES) is poor. Moreover,  for entail-edit datasets with large edit batch sizes (64 and 128), SEPROG clearly outperforms all the baselines by 13-20%.

---

### Decision · Program_Chairs · 2023-01-20

**Decision:**

Reject

**Justification For Why Not Higher Score:**

See weaknesses

**Justification For Why Not Lower Score:**

N/A

**Metareview: Summary, Strengths And Weaknesses:**

The authors propose a prompt generation method for reflecting model edits. Prompt vectors are generated based on the edits and provided to the model at inference time to reflect the new knowledge.

Strengths:
Most reviewers agreed that the authors were solving an important problem,

Weaknesses:
Most reviewers agreed that the method represented an incremental improvement over the SERAC method, which also used a semi-parametric approach for model editing. Also, they did not find the evaluations convincing enough.

I’m inclined to disagree with the reviewers that SEPROG is not a sufficient stand-alone contribution. While prior work in parameter-efficient tuning (e.g., prefix-lm) exists, this work has rarely been applied in a model editing setting, yielding sufficient novelty. However, I agree with the reviewers that the empirical evaluation could be stronger. In its current form, it’s not clear what the advantage of SEPROG would be. I think this paper has strong potential, but could use another round of iteration taking into account the feedback from the reviewers before a future acceptance.


**Summary Of Ac-Reviewer Meeting:**

N/A